# Enhancement of Methylene Blue Photodegradation Rate Using Laser Synthesized Ag-Doped ZnO Nanoparticles

**DOI:** 10.3390/nano12152677

**Published:** 2022-08-04

**Authors:** Damjan Blažeka, Rafaela Radičić, Dejan Maletić, Sanja Živković, Miloš Momčilović, Nikša Krstulović

**Affiliations:** 1Institute of Physics, Bijenička Cesta 46, 10000 Zagreb, Croatia; 2VINČA Institute of Nuclear Sciences—National Institute of the Republic of Serbia, University of Belgrade, P.O. Box 522, 11351 Belgrade, Serbia

**Keywords:** photocatalysis, ZnO nanoparticles, Ag-doped ZnO, zinc oxide, bicomponent nanoparticles, photodegradation, silver dopant, methylene blue, laser ablation in water, pulsed laser deposition

## Abstract

In this work, Ag-doped ZnO nanoparticles are obtained via pulsed laser ablation of the Ag-coated ZnO target in water. The ratio of Ag dopant in ZnO nanoparticles strongly depends on the thickness of the Ag layer at the ZnO target. Synthesized nanoparticles were characterized by XRD, XPS, SEM, EDS, ICP-OES, and UV–VIS spectrophotometry to obtain their crystal structure, elemental composition, morphology and size distribution, mass concentration, and optical properties, respectively. The photocatalytic studies showed photodegradation of methylene blue (MB) under UV irradiation. Different ratios of Ag dopant in ZnO nanoparticles influence the photodegradation rate. The ZnO nanoparticles doped with 0.32% silver show the most efficient photodegradation rate, with the chemical reaction constant of 0.0233 min^−1^. It exhibits an almost twice as large photodegradation rate compared to pure ZnO nanoparticles, showing the doping effect on the photocatalytic activity.

## 1. Introduction

Zinc oxide (ZnO) is a wide-bandgap semiconductor material with unique electrical, mechanical and optical properties, making it appropriate for a wide range of applications. Some of these properties are high optical transparency, high electron mobility (200–300 cm^2^ V^−^^1^ s^−^^1^), large direct bandgap (3.37 eV at room temperature), large exciton binding energy (60 meV), and high mechanical stability. ZnO is a low-cost, low-toxic, and easily available material, with many possible economical methods of synthesis that makes it appropriate for a broad range of scientific research and industrial applications [1]. ZnO is widely used in industry for the production of solar cells, light-emitting diodes and transistors, sensors, UV-protecting coatings, baby powders, vulcanization products, and for improving concrete and ceramic quality [2,3,4,5]. Photocatalytic, antibacterial, food-packaging, and medical (cancer-treatment) applications of ZnO are considered fast-developing technology [6,7,8]. The interest of this work is based on the photocatalytic properties of ZnO, especially those related to the photodegradation of surrounding chemicals in water, the process that can be used in water purification [9]. On the other hand, the widespread production and use of ZnO can lead to high concentrations of ZnO waste material in the aquatic environment that cause some degree of ecotoxicity, due to the large photocatalytic and antibacterial activity of ZnO [10]. In the photocatalytic process, electrons and holes, excited after light irradiation of semiconductor materials, interact with the surrounding medium, causing and accelerating many chemical and redox reactions. ZnO is the second most popular photocatalytic material after titanium oxide (TiO_2_), and in both materials, the processes that lead to photodegradation of the surrounding chemicals are similar [11]. Electrons in the conduction band transfer to O_2_, producing very reactive O^2−^ ions, while holes from the valence band transfer to hydroxide ion OH^−^, producing hydroxyl radical ·OH, which is very reactive [12]. The photocatalytic degradation of pollutants in water occurs due to interaction with reactive radicals. Light irradiation may also excite electrons in pollutants, leading to their transfer to the catalyst conductive band and the formation of additional reactive radicals [13]. The use of nanomaterials as photocatalysts is very efficient, due to the large area/volume ratio and better light irradiation harvesting. In nanoparticles, the same amount of material has a larger active site area and is irradiated with larger light intensity when compared to the bulk material. Both effects strongly increase the number of electrons and holes available for reactions at the catalyst surface. 

The wide bandgap of ZnO is the reason why UV light is needed for electron/hole excitation and this is a great limitation in photocatalytic applications because sunlight contains only 5% intensity of UV light in the whole EM spectrum [14]. However, the physical and chemical properties are easily adapted and modified by doping with various materials and surface engineering [12]. Many materials are used as dopants in ZnO to improve its photocatalytic activity and expand it to the visible part of the spectrum. Silver (Ag) seems to be the most promising doping material for this purpose [3]. Ag acts as a p-dopant in the ZnO crystal lattice, which in its pure form has a hexagonal wurtzite crystal structure that often has n-type conductivity, due to the presence of oxygen vacancies [13,15]. Doping with Ag is also used to improve the antibacterial activity of ZnO [16,17]. Furthermore, Ag, as a noble metal, can impose the photostability of ZnO, which is very important because of the fast photocorrosion, one of the main disadvantages of pure ZnO [12]. Doping can increase photocatalytic activity by creating band levels that act as traps for charge carriers. Therefore, the recombination rate of electrons and holes decreases, and their lifetime increases [14]. Several methods are usually applied for the synthesis of Ag-doped ZnO nanoparticles (ZnO NP). Some of them are photodeposition via photoreduction of Ag^2+^ [18,19], precipitation [20,21], the solvothermal route [14,22], using Ag nitrate in the sol-gel method [23], the hydrothermal route [24], laser induction [25], ultrasonic spray pyrolysis [26] and flame spray pyrolysis [27]. Pulsed laser ablation in liquid (PLAL) is a green method for the synthesis of colloidal solutions, which contains very pure nanoparticles without the unwanted residual byproducts [28]. Nanoparticles’ purity and the possibility of using a large variety of materials as a target in ablation are the main advantages of PLAL compared to chemical routes of NP synthesis. 

From the literature, it is evident that laser ablation in liquids has been applied for doping ZnO. In [29], Ag-doped ZnO nanoparticles were successfully synthesized by nanosecond laser ablation of Zn for 30 min and then Ag for 1, 3 or 5 min. Furthermore, some works were found where targets for laser ablation were prepared. The targets are produced through sintering at high temperatures and compressing, which can last up to two days and has many steps in the production process [30,31,32]. 

In our work, we propose a new method, which is thoroughly described in the experimental section, for the production of Ag-doped ZnO NPs. An Ag thin layer is deposited at the ZnO target surface with the pulsed laser deposition (PLD) technique, and the obtained target is ablated by a pulsed laser in water. As a result, a colloidal solution of Ag-doped ZnO nanoparticles is formed. Furthermore, a ratio of Ag dopant in the ZnO NP is varied by using different thicknesses of Ag layer at the ZnO target irradiated in PLAL. ZnO NPs, with different ratios of Ag dopant, are then tested as a catalyst in photocatalytic degradation of methylene blue. The method is simple, fast and versatile, allowing a combination of a wide variety of dopants and semiconductors. Moreover, the overall process (including PLD and PLAL) for the synthesis of Ag-doped ZnO NP colloidal solution takes approximately 10 min.

## 2. Materials and Methods

### 2.1. Preparation of the Ag Thin Films at the ZnO Substrates by PLD 

The thin films of Ag were deposited at the ZnO substrates (purity > 99.99%, GoodFellow, Huntingdon, UK) by PLD in a vacuum. The 1 mm silver plate (purity > 99.99%, GoodFellow) was used as the laser target in PLD. The ZnO substrate and the Ag target were placed at the two stages separated 1 cm inside the vacuum chamber, which both rotated to ensure the homogenous laser irradiation of the Ag target and homogeneity of the deposited Ag film. An Nd:YAG laser (300 mJ, 5 Hz, 5 ns pulse duration, wavelength 1064 nm, Quantel, Brilliant, Les Ulis, France) was used for Ag target irradiation in the PLD process. Laser pulses were focused onto the target surface using a 30 cm lens yielding laser fluence of 12 J/cm^2^. In order to make samples with different thicknesses of the silver film at the ZnO substrate, the following six different numbers of pulses were used in PLD: 50, 200, 500, 1000, 2000, and 3000. The detailed schematics procedure of the two-step synthesis of Ag-doped ZnO nanoparticles is shown in Figure 1 [33]. As a first step, in Figure 1a, pulsed laser deposition of Ag onto ZnO substrate is shown. As a result of the deposition, a two-layer structure was obtained, consisting of several nanometers thick Ag thin film deposited over the ZnO substrate, as shown in Figure 1b.

### 2.2. Syntheses of the Ag-Doped ZnO NP Using PLAL

In the second step, an Ag-coated target was used for the pulsed laser ablation in the water process, producing an Ag-doped ZnO colloidal solution as a final product (Figure 1d). The colloidal solutions of nanoparticles were prepared for six targets, synthesized by PLD in (Figure 1a) to obtain the ZnO NPs with different Ag dopant ratios. The pure ZnO target was ablated to synthesize the colloidal solution of pure ZnO NP as a reference solution. The target, as shown in Figure 1b, was immersed in the 25 mL of MilliQ water and placed about 2.5 cm under the water surface. The surface of approximately 1 cm^2^ in size was continually scanned by the pulsed laser beam to achieve homogenous ablation of the sample. The total number of pulses for irradiation of each sample was 2000, using the same laser as for PLD. The laser beam was focused by a 10 cm lens to enhance ablation. The energy delivered to the sample was about 120 mJ per pulse, and the calculated fluence was 79 J/cm^2^. The experimental setup is shown in Appendix A.

### 2.3. Characterization of the Ag-Doped ZnO NP

#### 2.3.1. UV–Vis

Photoabsorbance of every colloidal solution of Ag-doped or pure ZnO NP was measured in order to analyze the influence of Ag doping on ZnO NP optical properties. Photoabsorbance data were used to determine the ZnO NP bandgap from the Tauc plot. Photoabsorbance was measured immediately after the synthesis of every colloidal solution of nanoparticles, using a UV–Vis spectrophotometer (Lambda 25, Perkin Elmer, Waltham, MA, USA).

#### 2.3.2. Structural Analysis (XPS, XRD, SEM, TEM)

In order to perform structural characterization of ZnO NPs, the produced colloid was dropped onto a silicon substrate and left to air dry to obtain the ZnO NP film.

The X-ray photoelectron spectroscopy (XPS) analysis was carried out with the PHI-TFA XPS spectrometer produced by Physical Electronics Inc. The analyzed area was 0.4 mm in diameter and the analyzed depth was about 3–5 nm. The sample surface was excited by X-ray radiation from a monochromatic Al source at photon energy of 1486.6 eV. The high-energy resolution spectrum was acquired with an energy analyzer, operating at a resolution of about 0.6 eV and pass energy of 29 eV. During data processing, the spectrum was aligned by setting the C 1s peak at 284.8 eV, characteristic for the C-C/C-H bonds. The accuracy of binding energies was about ±0.3 eV. Quantification of surface composition was performed from XPS peak intensities, considering the relative sensitivity factors provided by the instrument manufacturer [34]. Two different XPS measurements were performed on each sample, and the average composition was calculated.

The crystallinity and crystalline phases were studied by grazing incidence X-ray diffraction (GIXRD). The crystalline structure of ZnO NP was investigated using a D5000 diffractometer (Siemens, Karlsruhe, Germany) in parallel beam geometry with Cu Kα radiation, a point detector, and a collimator in front of the detector. Grazing incidence X-ray diffraction (GIXRD) scans were acquired with the constant incidence angle α_i_ of 1°, ensuring that the information contained in the collected signal covered the entire film thickness. The following JCPDS cards were used for crystallographic analysis: JCPDS 36-1451 (ZnO), JCPDS 04-0783 (Ag), JCPDS 43-1038 (AgO), and JCPDS 76-1393 (Ag_2_O). 

The morphology and size-distribution of the obtained nanoparticles were studied by a field emission scanning electron microscope (SEM, Joel 7600F, Tokyo, Japan). Samples for SEM imaging were prepared by dropping a colloidal solution on a 5 × 5 mm^2^ Si wafer. The colloidal solutions were sonicated for 5 min before the deposition. 

The detailed morphology of the nanoparticles was obtained by a transmission electron microscope (TEM, JEOL JEM-1400 Flash, Tokyo, Japan). Samples for imaging were prepared by dropping one drop of colloidal solution onto the TEM grid. The colloidal solutions were sonicated for 5 min before the deposition. 

#### 2.3.3. ICP-OES

Quantitative elemental analysis of the nanoparticles, prepared in the form of aqueous solutions, was performed using a Thermo Scientific iCAP 7400 duo analyzer, an instrument used for inductively coupled plasma optical emission spectrometry (ICP-OES). ICP-OES measurements were taken to calculate the mass concentration of the ablated material and to estimate the Ag weight ratio in the colloidal particles. Before analysis, 2mL of each NP solution was treated with the same volume of analytical grade 65% (*w*/*v*) HNO_3_ or aqua regia (Au, Pt), and diluted to 10 mL with distilled high-purity deionized water in a volumetric flask. The calibration standard solutions of Zn, Au, Pt, and Ag were prepared from a single element commercial standard (J.T. Baker 1000 µg/mL and Accu Trace (1000 μg/mL)) by diluting it to different volumes of distilled high-purity deionized water with 1% (*w*/*v*) analytical grade HNO_3_. The data acquisition and processing stages were performed by using the Thermo Scientific Qtegra Intelligent Scientific Data Solution (ISDS, Waltham, MA USA) software. The obtained results are presented in the form of the average values of the three measurements.

### 2.4. Photocatalytic Measurements

An Hg lamp (Jelight Co., Zagreb, Croatia) was used for irradiation of the solutions that contained the pure and Ag-doped ZnO NPs and the organic dye methylene blue was used to achieve MB photocatalytic degradation. The irradiation intensities of the lamp were measured using a radiometer (Opsytec Radiometer RMD, Ettlingen, Germany) and the intensities were as follows: VIS 2.11 klx, UVA 0.2 mW/cm^2^, UVB 0.02 mW/cm^2^ and UVC 0.08 mW/cm^2^. The optical emission spectrum of the lamp is shown in Appendix A. For purposes of MB photodegradation rate calculation, solution absorbance was measured every 10 min of irradiation using a UV–Vis spectrophotometer. Initial MB concentration for all the samples corresponds to the absorbance maxima A_0_ = 2 of the pure MB solutions in water and was calculated to be 2.7 × 10^−^^5^ mol/L from the Beer–Lambert law, using the extinction coefficient 75,000 cm^−^^1^/M and A_0_ = 2 at peak maximum at λ = 664 nm.

## 3. Results and Discussion

### 3.1. SEM and TEM Images

The SEM scans of all the samples show that ablation of the ZnO target resulted in the formation of colloidal particles, which were distributed in a large variety of sizes and shapes. It is worth mentioning that morphology and size-distribution of the nanoparticles is independent from the amount of silver that differs by a few nanometers (including pure ZnO NP). In Figure 2, examples of the SEM micrographs are shown for the pure and Ag-doped ZnO NP. In Figure 2a,c, it can be observed that the spherical nanoparticles prevail, and some small amounts of microparticles in the form of large debris structures with irregular shapes are present. The size-distributions of pure and Ag-doped ZnO NP are shown in Figure 2b,d, respectively. Size-distributions can be fitted with LogNormal functions that achieve a maximum value at a diameter of 53 nm for pure and 47 for Ag-doped ZnO NPs.

In Figure 3, TEM images for pure (a, b) and Ag-doped (c, d) ZnO NP are presented. Figure 3a,c show the presence of spherical nanoparticles with different sizes and irregular material formed during the LAL process. Figure 3b,d show the surface morphology of the nanoparticles. It can be observed that in both cases, the NP surface is relatively smooth. Nevertheless, the edges are not perfectly sharp, indicating that the NP surface has some roughness. The differences in photocatalytic activities between the pure and Ag-doped ZnO NPs are obviously not dependent on surface morphology.

### 3.2. XRD Patterns

The XRD spectrum of colloidal Ag-doped ZnO nanoparticles (3000p used for Ag deposition in PLD) and reference XRD spectral peaks for ZnO, Ag, AgO, and Ag_2_O are shown in Figure 4. The majority of the detectable Bragg peaks (31.75°-(100), 34.45°-(002), 36.2°-(101), 47.6°-(102), 56.55°-(110), 62.85°-(103), 68°-(112) and 69.1°-(201)) correspond to ZnO. The majority of the ablated material crystallized in the hexagonal wurtzite ZnO phase. The presence of all main ZnO diffraction peaks indicates that ZnO crystallites are randomly oriented. The low-intensity peak at 32.75° can probably be attributed to the corresponding peak in the AgO reference spectrum. The absence of Ag peaks indicates that Ag in the colloidal solution is not in the form of pure Ag nanoparticles or Ag crystalline clusters in ZnO nanoparticles, indicating that Ag atoms are incorporated in the ZnO crystal lattice. Since the sample selected for XRD analysis is synthesized by using the largest number of laser pulses in PLD (3000p), it is expected that in all other samples, the ZnO wurtzite crystal structure and some amount of Ag also prevail, which is well-incorporated in the ZnO crystal lattice.

However, to study the effect of Ag doping on the ZnO lattice diffraction peaks for (100), (002), and (101) planes were compared for the doped and pure ZnO nanoparticles, as is shown in Figure 5.

The XRD peaks of the Ag-doped ZnO NPs shifted to a lower 2*θ* angle value, as compared with those of pure ZnO nanoparticles. This shift appears due to the fact that the radius of the incorporated Ag^+^ ions is greater than that of Zn^2+^, which consequently causes lattice expansion [35]. The FWHM of the pure ZnO diffraction peaks is narrower than the Ag-doped ZnO peaks, implying that the doping causes the formation of smaller crystallites [36]. The average crystallite size (*D*) was calculated using Debye–Scherrer’s formula [37], using data from the pure and Ag-doped ZnO main XRD peaks.
(1)$$D=\frac{0.9\cdot \lambda }{\beta \cdot \cos\theta }$$
where *λ* is the wavelength of the X-ray beam (1.5406 Å), *β* is the full width at half maximum (in radians), and *θ* is the diffraction or Bragg’s angle. The calculated average crystallite sizes are 49 ± 6 nm for the pure ZnO and 23 ± 1 nm for the Ag-doped ZnO. Furthermore, one must note that the relative intensities of all the diffraction peaks do not change significantly for the Ag-doped and pure ZnO structures. It means that in both cases, there is no preferential growth rate in any direction and that the presence of Ag atoms/ions does not change the growth mechanism of the ZnO nanoparticles in LAL, as described in [38]. The Ag doping causes lattice structural changes, which are also evident if one compares a and c lattice constants for doped and pure samples of the wurtzite ZnO structure. The lattice constants are calculated using Bragg’s law of diffraction, which is as follows:(2)n·λ=2·d·sin(θ)
where *n* = 1 (order of diffraction), *λ* is the X-ray wavelength (*λ* = 1.5406 Å), *d* is the spacing between the planes defined by Miller indices (h,k,l), and *θ* is the position of the diffraction peak. The relation of plane spacing in a hexagonal structure and Miller indices are defined in [39], while ((100) orientation) and *c* ((002) orientation) spacing can be calculated using the following formulae:(3)$$a=\frac{\lambda }{\sqrt{3}\cdot sin\;\theta }$$
(4)$$c=\frac{\lambda }{sin\;\theta \;}$$

The lattice constants for Ag-doped and pure ZnO nanoparticles are listed in Table 1.

From Table 1, it is evident that the lattice constants a and c are larger for Ag-doped than that of pure ZnO nanoparticles, implying lattice expansion. An increase in lattice parameters can be ascribed to the replacement of Zn^2+^ with Ag^+^ ions, as they possess a larger ionic radius (radius is 0.074 nm for Zn^2+^ ions and 0.126 for Ag^+^ ions). As discussed in [39], an increase in lattice constants can be achieved either by the incorporation of Ag^+^ ions into the ZnO lattice or by the substitution of Ag^+^ ions with Zn^2+^ ions. 

### 3.3. XPS Analysis

The XPS technique was applied to obtain an insight into the surface composition of Ag-doped ZnO nanoparticles. In Figure 6, the full XPS spectrum of Ag-doped ZnO NP is presented for a sample where 3000 pulses were used for Ag deposition in PLD. The presence of characteristic peaks can be identified for zinc (Zn 2p, Zn 3s, Zn 3p, Zn 3d, and Auger peaks Zn LMM), oxygen (O 1s and Auger peak O KLL), silver (Ag 3d, shown in inset) and carbon (C 1s). The spectrum indicates that Ag-doped ZnO nanoparticles are successfully synthesized.

The XPS high-resolution spectra are shown in Figure 7 for the following peaks: (a) Zn 2p_3/2_, (b) Ag 3d, (c) O 1s, and (d) C 1s. Deconvolution of the Zn 2p_3/2_ peak (Figure 7a) reveals that a gaussian distribution with a maximum at 1021.5 eV, which corresponds to Zn^2+^ states in the ZnO crystal lattice [40], is dominant. The origin of the gaussian distribution with a maximum at 1019.7 eV is unclear, but its influence is small. Among the Ag 3d states (Figure 7b), the Ag 3d_5/2_ peak at 367.3 eV and Ag 3d_3/2_ peak at 373.3 eV are present, which are probably related to Ag-O bonds, because the same binding energies are characteristic for Ag_2_O [41]. The expected binding energies of the Ag 3d state for pure metallic silver are 368.3 eV for Ag 3d_5/2_ and 374.3 eV za Ag 3d_5/2_, while their absence, as is the case here, points to the conclusion that silver appears not in the form of metallic Ag nanoparticles but is incorporated in ZnO. Deconvolution of the O 1s (Figure 7c) peak results in two gaussian fits, first with a maximum at 529.9 eV, related to oxygen that builds up in the ZnO crystal lattice, and second with a maximum at 531.6 eV that is related to hydroxyl (O-H) oxygen [39]. Deconvolution of C 1s states (Figure 7d) results also in two gaussian fits, first with a maximum at 288.6 eV, related to C=O bonds, and second with a maximum of 284.7 eV, related to C-C bonds [42]. Carbon occurs as a sample impurity.

XPS spectral data are used for the atomic ratio calculation for each element present in the XPS spectrum (Zn, O, Ag, and C). From atomic ratios, after subtraction of impurities, the Ag weight ratio in ZnO NP can be calculated, but such calculated value pertains only to the depth range 3–5 nm monitored by XPS. In Figure 8, the Ag weight ratio (calculated using XPS data), which is dependent on the number of pulses used for Ag deposition in PLD, is shown. It can be observed that the Ag silver ratio grows while the PLD number of pulses increases, up to a maximal value of 4.5 wt.% at 3000 pulses.

### 3.4. ICP-OES Measurements

ICP-OES measurements revealed that Zn mass concentration in each colloidal solution is approximately the same and has the value of 80 ± 3 mg/L. Under the assumption that the majority of synthesized colloidal material is Ag-doped ZnO, the mass concentration of ZnO is calculated to be 101 ± 3 mg/L. The Ag weight ratio in colloidal particles is estimated from the ICP-OES results, under the assumption (justified from XRD and XPS analysis) that the majority of Ag atoms are incorporated in ZnO. Figure 9 shows the ICP-OES calculated Ag weight ratio in Ag-doped ZnO NPs for a different number of pulses used for Ag deposition during PLD. It can be observed that the Ag weight ratios calculated from ICP-OES (Figure 9) are much smaller than those obtained from XPS (Figure 8). For instance, in the sample where 3000 pulses were used for Ag deposition in PLD, the Ag weight ratio calculated from XPS is 4.5%, while ICP-OES gives 0.55%, which is 8 times smaller. In the ICP-OES measurements, all colloidal material from the solution is included, while XPS monitors only in the 3–5 nm depth range and only at specific points on the samples. The Ag weight ratios obtained from ICP-OES have much higher accuracy, but large XPS values point to the conclusion that Ag atoms are mostly concentrated close to the ZnO surface.

In Figure 9, it can be observed that the Ag weight ratio growth slows down with the increasing number of pulses. Such a relation is probably related to the similar dependence of a number of deposited Ag atoms at the ZnO substrate on a number of laser pulses during PLD. There are a few possible explanations for such a relation. First, incident Ag atoms from the plasma plume formed during ablation in the PLD process could have enough energy to induce sputtering of other Ag atoms that are already deposited on the ZnO substrate. Secondly, Ag ions are positively charged, so the deposited Ag film could also be positively charged because the ZnO substrate lays on an ungrounded base. Therefore, incident Ag ions could be electrically repulsed from the Ag film, and this repulsion is expected to be larger when more Ag atoms are already present in the Ag film. The deposition rate is expected to decrease while increasing the number of pulses in PLD. Thirdly, by increasing the number of pulses in PLD, the average number of pulses that hits the same position of the Ag target also grows. It means that the focusing conditions deteriorate with irradiation time, so the same energy of the laser beam is distributed over a larger area of the Ag target. Therefore, single plasma species created during ablation have lower average energy, which means that a smaller number of Ag atoms have sufficient energy for reaching the ZnO substrate.

### 3.5. Photoabsorption Measurements

In Figure 10, photoabsorbance measurements are shown for the colloidal solutions of pure ZnO NP and Ag-doped ZnO NP synthesized by ablation of the ZnO target, covered with an Ag layer deposited with 1000 pulses in PLD. The UV–Vis spectra differ in intensity, where the Ag-doped ZnO colloidal solution has higher absorbance in the visible part of the spectrum and lower absorbance in the UV part concerning absorbances for pure ZnO NP. The total mass of Zn measured by ICP-OES is 79 mg/L and 85 mg/L for Ag-doped and pure ZnO, respectively. From the similarity of those masses, it can be concluded that a similar amount of material is involved in photocatalysis. The UV–Vis spectra differ either since Ag-doped ZnO NPs are larger than pure ZnO NPs, which was not found in the SEM observations (it applies to all samples), or that in the Ag-doped ZnO NP solution, more material crystallizes in irregular shaped structures, which is hard to quantify from SEM images. One must note that the metallic Ag peak with a maximum at about 400 nm [43] is not visible, and this further confirms that Ag is incorporated in the ZnO crystal lattice, rather than appearing in the form of pure Ag NPs.

Photoabsorbance measurements of the as-synthesized Ag-doped ZnO NP colloidal solutions (as shown in Figure 10) were used for bandgap calculations. Tauc plot for direct bandgap calculation ((Ahν)^2^ vs. hν) was used to determine the bandgap of pure and Ag-doped ZnO NPs. Tauc plots for pure ZnO NP and Ag-doped ZnO NPs (produced with 1000 pulses in PLD) are shown in the inset of Figure 10. The direct bandgap is calculated as the intercept on the x-axis of the linear part of the Tauc curve and has a value of 3.27 eV for pure and 3.21 eV for Ag-doped ZnO. It was found that the bandgap energies gradually drop from 3.27 eV for pure to 3.14 eV for Ag-doped ZnO produced with 3000 pulses in PLD. It is evident that the bandgap energy decreases when the Ag weight ratio increases. It implies that the incorporated Ag ions influence the narrowing of the bandgap of ZnO, extending the absorption to the visible part of the spectrum, and thus leading to more efficient absorption of light. 

### 3.6. Photocatalytic Efficiency Determination

In order to measure the photocatalytic efficiency of Ag-doped ZnO NPs, the MB photodegradation process during UV irradiation was monitored for 1 hour in 10 min steps for each sample. In Figure 11, the example of a gradual decrease in the photoabsorbance curve during irradiation time is shown. In this example, ZnO NPs are synthesized by using 1000 Ag pulses in PLD. The photoabsorbance maximum decreased from 2.15 to 0.60 (or by 73%) during 60 min of irradiation, which points to the relatively strong photocatalytic activity of Ag-doped ZnO NPs. 

The photocatalytic activity of Ag-doped ZnO NPs is based on the excitation of the electrons from the valence band (VB) to the conduction band (CB) under light irradiation that is equal or higher than the band-gap energy. When electrons are excited, holes are formed in the VB that act as positive charge carriers. The unique Ag/ZnO Fermi level is below the bottom of the ZnO CB, so photo-excited electrons flow from ZnO to the Ag surface (Ag is an “electron sink”), while the holes remain on the ZnO surface. Moreover, silver doping can increase photocatalytic activity by creating band levels that act as traps for charge carriers. Therefore, the recombination rate of electrons and holes decreases, and their lifetime increases [14]. New band levels can also lead to effective band-gap narrowing that expands the range of photon energies, which can lead to photocatalytic efficiency in the visible part of the spectrum [12]. Silver, as a plasmonic metal, strongly enhances the light absorption via surface plasmon resonance (SPR), increasing the number of excited electrons and holes in the catalyst, consequently leading to higher photocatalytic performance [12]. Ag usually has a higher value of work function (4.3 eV [44]) than the absolute value of ZnO electron affinity (about 4.1 eV [45]). When the balance between the Fermi levels is established, the Schottky barrier is formed at ZnO/Ag [18]. The Schottky junction is the depleted area where a strong electric field separates the electrons and holes. This process decreases the e−/h+ recombination rate and, therefore, improves photocatalysis. In photocatalytic reactions, electrons in the CB react with O_2_, generating superoxide anions (·O_2_^−^), while holes in the VB react with H_2_O, generating hydroxyl radicals (·OH). Therefore, these free radicals participate in the photodegradation of organic dyes, such as MB [46,47,48]. In [48], Ag-doped ZnO nanocatalysts were synthesized using the co-precipitation method. Intriguingly, a scavenger test was carried out to determine the radicals that affect MB photodegradation. The results showed that photo-generated holes and ·OH radicals are the main species, while ·O_2_^−^ radicals are the supportive species in the photodegradation process of MB. We assume that the same radicals dominate in our case, as the crystallinity of the nanoparticles and Ag weight ratios are similar. 

The photodegradation rates (*k_DR_*) were calculated by the following equation, assuming that the reaction follows pseudo-first-order kinetics:(5)$$C=C_{0}\cdot e^{-k_{DR}\cdot t}$$
where *C* is a concentration of MB at time *t*, *C*_0_ is an initial MB concentration, *k_DR_* is the degradation rate, and *t* is the irradiation time. In Figure 12, the ln(*C*/*C*_0_) vs. *t* plot is shown for each sample, where *C*/*C*_0_ is the ratio between the time-dependent concentration *C* of MB and initial concentration *C*_0_ of MB. *C* and *C*_0_ are, according to the Beer–Lambert law, proportional to the corresponding values of photoabsorbance maxima, from which the photoabsorbance of the solution without MB (zero-curve) is distracted, so *C*/*C*_0_ is simply calculated as the ratio between these photoabsorbance values. Photodegradation rate *k_DR_* is calculated as the slope of the linear fit in ln(*C*/*C*_0_) vs. *t* plot for each case.

Table 2 summarizes the Ag weight ratios in Ag-doped ZnO NPs calculated from ICP-OES, the photodegradation rates, and corresponding photodegradation half-times for each irradiated colloidal solution.

In Figure 13, the dependence of the photodegradation rate on Ag weight ratio in ZnO NPs is shown. The photodegradation rate for the pure ZnO NP colloidal solution is 0.0126 min^−^^1^. As we introduce Ag dopant and increase its mass concentration in the nanoparticles, the photodegradation rate increases. It reaches a maximum value of 0.0233 min^−^^1^ for 0.32 Ag wt.% (1000p in PLD), which is almost the same as 0.230 min^−^^1^, the value achieved for 0.28 Ag wt.% (500p in PLD). For the higher concentrations of Ag (2000 and 3000 pulses in PLD), the photodegradation rate decreases and drops to the value of 0.0101 min^−^^1^ for 0.55 Ag wt.% (3000p in PLD), which is lower than the photodegradation rate of pure ZnO NP. The existence of an optimal Ag dopant ratio for maximization of photocatalytic activity is observed in numerous papers [13,49,50,51,52,53,54] and mostly has a value between 0.5 wt.% and 2.5 wt.%. The growth of *k_DR,_* while increasing the Ag ratio in ZnO NP, can be explained as an interplay of the Ag doping effects, which are favorable for photocatalysis, such as decreased recombination rate due to induced Ag intraband levels, improved charge separation, bandgap narrowing, and the SPR effect. The decrease in *k_DR_* in the Ag dopant ratios that are larger than the optimal ratio can be caused by several effects. For example, defects induced in the ZnO crystal lattice, due to the presence of the Ag material, can act as recombination centers for electron–hole pairs and decrease their lifetime [20]. This is attributed to the electrostatic attraction of positively charged holes and negatively charged Ag. Other reasons include the decreased available area for UV light absorption and decreased reactant adsorption due to the presence of Ag at the ZnO surface [27]. This leads to the decrease in the photons absorbed by ZnO NP, lowering the photocatalytic reaction efficiency. Furthermore, decrease in the holes reacting with organic pollutants at the ZnO surface can occur. As was concluded from the comparison of the Ag ratios calculated by ICP-OES and XPS, the Ag atoms are probably found mostly close to the surface of the ZnO catalyst. Their contribution to all the mentioned effects, either beneficial or detrimental for photocatalysis, is larger than in the case of homogenous doping. The reason is that both light irradiation and photocatalytic interactions occur at the catalyst surface. In addition, this is probably the reason why the optimal Ag weight ratio for maximization of photocatalytic activity, calculated from ICP-OES, which considers the whole volume of the catalyst material, is relatively low (0.32 wt.%). The large decrease in *k_DR_* from 0.0233 min^−^^1^ to 0.0169 min^−^^1^, which occurs while increasing the silver weight ratio by a very small amount (from 0.32% to 0.36%, ICP-OES), can be explained by the fact that at the same time, the silver weight ratio close to the ZnO surface drastically increased (from 1.3% to 4.1%, XPS).

It is interesting to compare the obtained results with results in similar works [21,50,51,54]. In [50], 95.6% photodegradation efficiency of amaranth dye was achieved after 4 h with the addition of 1% of Ag to ZnO, but with a total amount of catalyst of 160 mg/L. In our case, 73% of MB photodegradation after one hour was achieved with a catalyst amount of around 100 mg/L. These two cases cannot be directly compared but can serve as a point to assess the efficiency of photocatalysis presented in this work. In other papers, the amount of catalyst is not reported, so a direct comparison with our work cannot be made. In [51], the photocatalytic behavior of nanostructured sol-gel synthesized Ag-doped ZnO films for degradation of MB under UV irradiation was studied. The optimal weight ratio of the Ag dopant was 0.5%, with an MB photodegradation rate that was 1.6 times larger (0.039 min^−^^1^) when compared with pure ZnO (0.024 min^−^^1^). This is a slightly lower improvement than that achieved in our work (2 times larger *k_DR_*), while comparing the cases with optimal Ag doping and pure ZnO. In [54], Ag-doped ZnO NPs were synthesized by combining the co-precipitation method and HILF (hybrid induction and laser heating) technique and their photocatalytic efficiency was analyzed for MB degradation under UV irradiation. The largest MB photodegradation rate (0.126 min^−^^1^–85% in 15 min) was achieved for the Ag/Zn ratio of 2%, which corresponds to the Ag weight ratio of 2.5 wt.%. It is about 5 times larger than the one obtained for pure ZnO NPs (0.024 min^−^^1^–30% in 15 min). In [21], Ag-doped ZnO NPs were synthesized by the co-precipitation method and photodegradation of brilliant green dye under UV irradiation was examined in their presence. Optimal photocatalytic efficiency (0.031 min^−^^1^) was obtained at the Ag weight ratio of 2%, and it is 2.6 times better than the one obtained for pure ZnO (0.012 min^−^^1^). Here, the strong bandgap narrowing is also noticed, from 3.09 eV for pure ZnO to 2.79 eV for 2% Ag-doped ZnO. Although optimal Ag doping in papers [21,54] increased the photodegradation rate of ZnO more than in the case of the present paper (5 and 2.6 compared to 2), it should be noted that in these papers, it is achieved at much larger Ag weight ratios (2.5 wt.% and 2 wt.% compared to 0.32%). A low optimal Ag ratio, such as that achieved in this paper, can be an advantage from the cost-effectivity point of view.

## 4. Conclusions

Ag-doped ZnO NPs are successfully synthesized using the combination of PLD and PLAL laser methods. The difference in silver weight ratio calculated by XPS and ICP-OES points to the conclusion that the silver atoms prevail close to the ZnO NP surface, and therefore have a greater impact on ZnO photocatalytic efficiency than in the case of homogenous doping. The largest photocatalytic efficiency is obtained by using ZnO NPs doped with 0.32 wt.% silver, with an MB photodegradation rate that is twice as large as that obtained in the presence of pure ZnO NP. The photocatalytic efficiency is improved by silver doping, due to its favorable impacts on photocatalysis, such as decreasing the recombination rate caused by inducing energy levels in the ZnO band gap, which acts as charge carrier traps and the activity of ZnO is evident, implying good effectivity of silver as a doping material. 

## 5. Patents

Croatian patent application involves the method of production of two-component nanoparticles using a laser (patent application number P20211098A).

## Figures and Tables

**Figure 1 nanomaterials-12-02677-f001:**
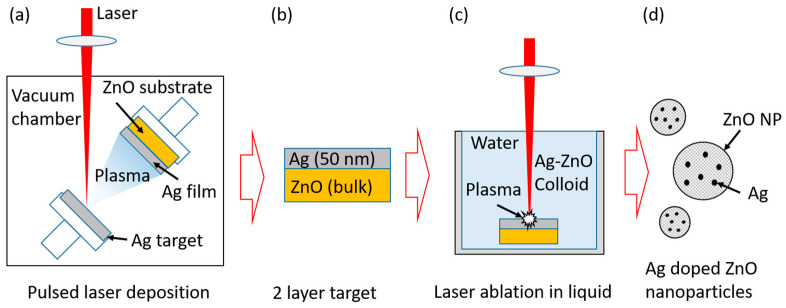
Experimental scheme for the synthesis of Ag-doped ZnO nanoparticles. (**a**) Pulsed laser deposition of Ag onto ZnO substrate (bulk), (**b**) two-layer target formed in PLD comprised of thin Ag film and ZnO bulk, (**c**) laser ablation of two-layer target in water and (**d**) Ag-doped ZnO nanoparticles synthesized in (**c**).

**Figure 2 nanomaterials-12-02677-f002:**
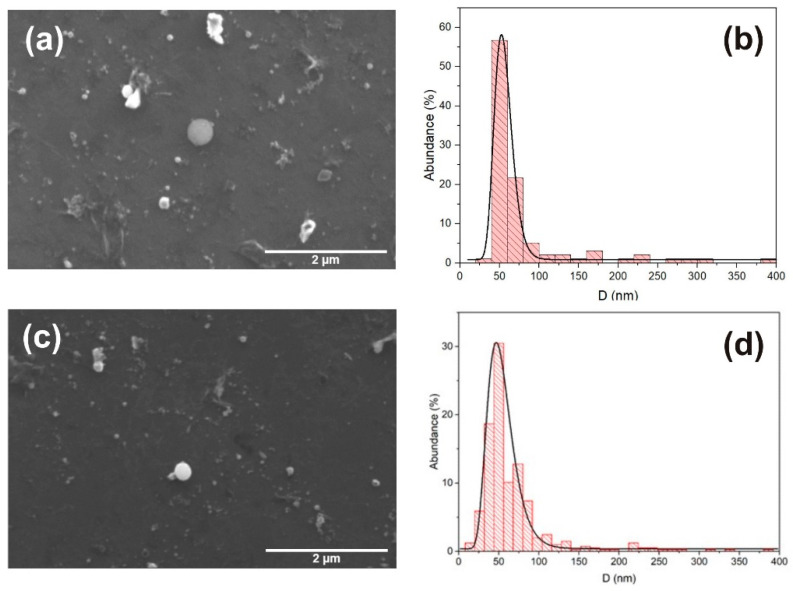
SEM micrograph of (**a**) pure ZnO NP with (**b**) corresponding size-distribution with maximum at 53 nm. SEM micrograph of (**c**) Ag-doped ZnO NP with (**d**) corresponding size-distribution with maximum at 47 nm.

**Figure 3 nanomaterials-12-02677-f003:**
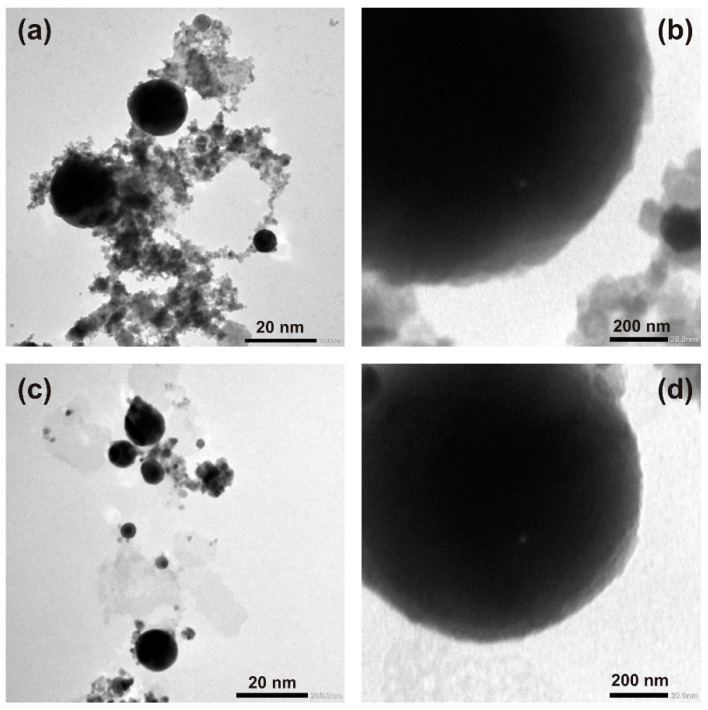
TEM images of pure ZnO NP (**a**,**b**) and Ag-doped ZnO NP (**c**,**d**).

**Figure 4 nanomaterials-12-02677-f004:**
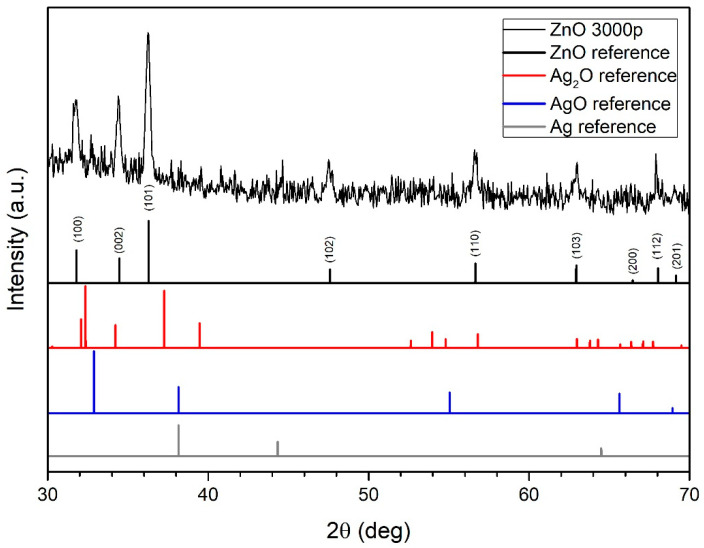
XRD spectrum of ZnO NP (3000 pulses used for Ag deposition in PLD) and reference XRD spectral peaks of ZnO, Ag, AgO and Ag_2_O.

**Figure 5 nanomaterials-12-02677-f005:**
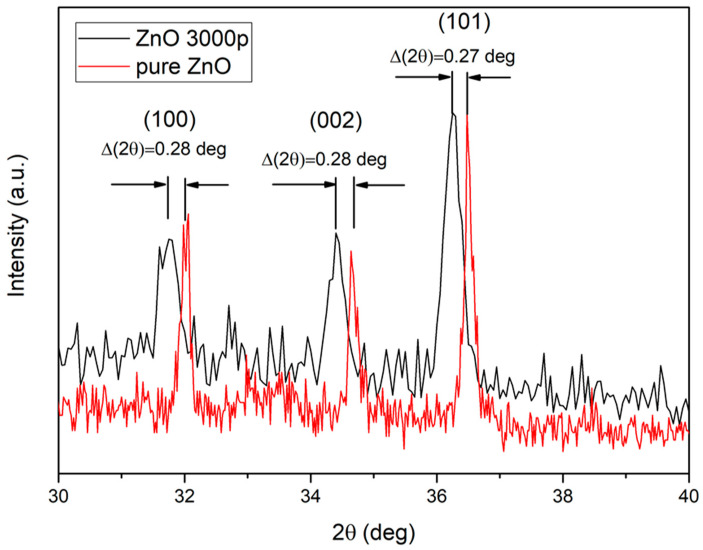
The shift of Ag-doped ZnO XRD peaks, with respect to the pure ZnO peaks.

**Figure 6 nanomaterials-12-02677-f006:**
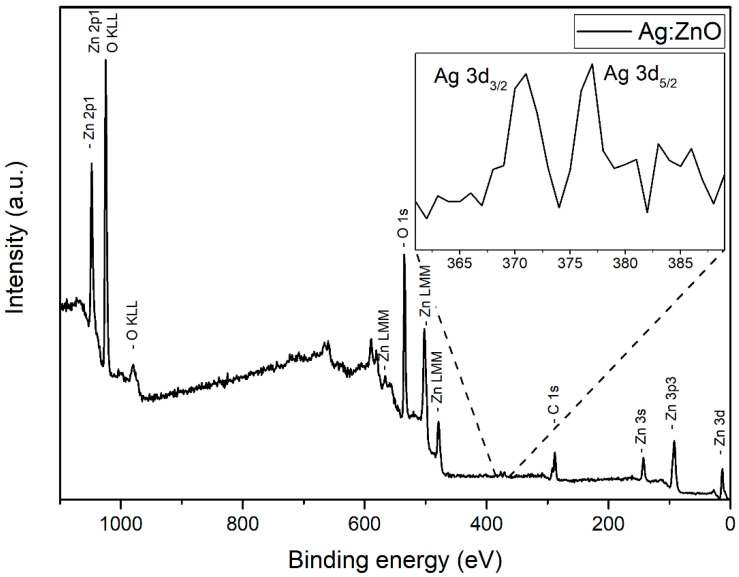
XPS spectrum of Ag-doped ZnO NP (3000 pulses used for Ag deposition in PLD).

**Figure 7 nanomaterials-12-02677-f007:**
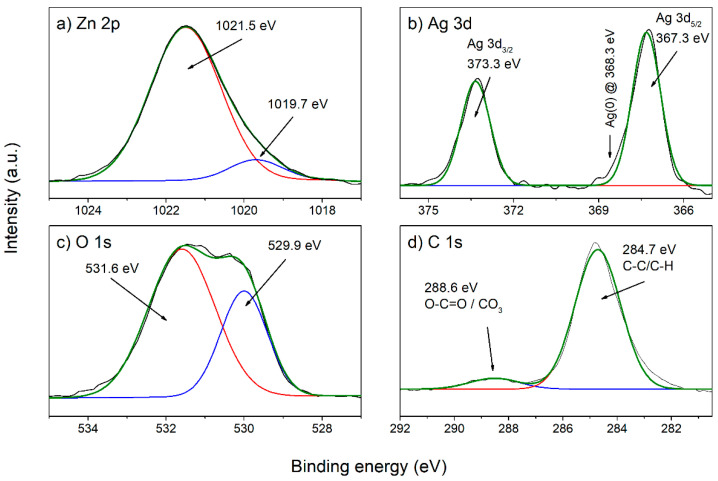
High-resolution spectra of Ag-doped ZnO NP (3000p used for Ag deposition in PLD) with fit spectra for (**a**) Zn 2p_3/2_, (**b**) Ag 3d, (**c**) O 1s and (**d**) C 1s.

**Figure 8 nanomaterials-12-02677-f008:**
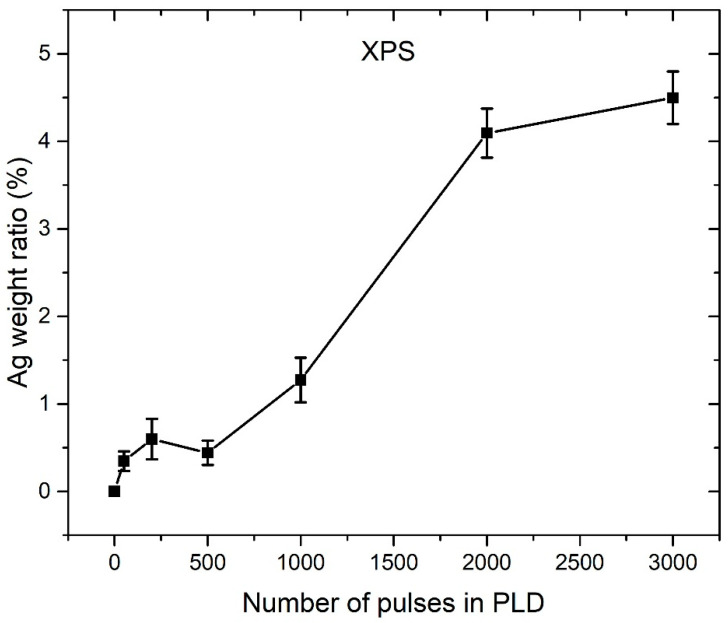
XPS calculated Ag weight ratio in PLAL-synthesized Ag-doped ZnO NP, which is dependent on number of pulses used for Ag deposition in PLD synthesis of PLAL target.

**Figure 9 nanomaterials-12-02677-f009:**
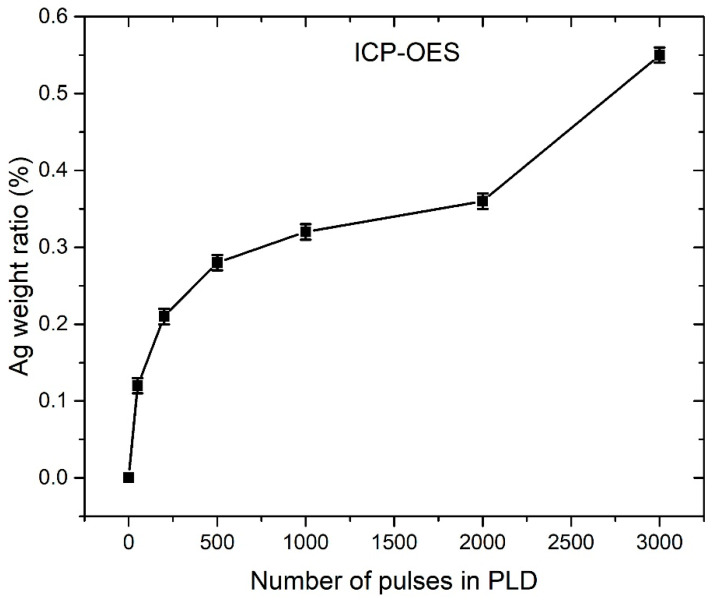
ICP-OES calculated Ag weight ratio in PLAL-synthesized Ag-doped ZnO NP that is dependent on number of pulses used for Ag deposition in PLD synthesis of PLAL target.

**Figure 10 nanomaterials-12-02677-f010:**
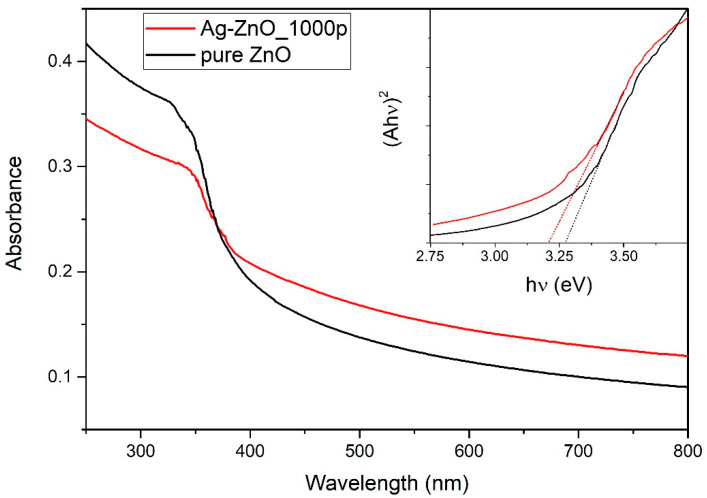
Photoabsorbance spectrum of colloidal solutions of pure ZnO NPs and Ag-doped ZnO NPs synthesized with 1000 pulses in PLD.

**Figure 11 nanomaterials-12-02677-f011:**
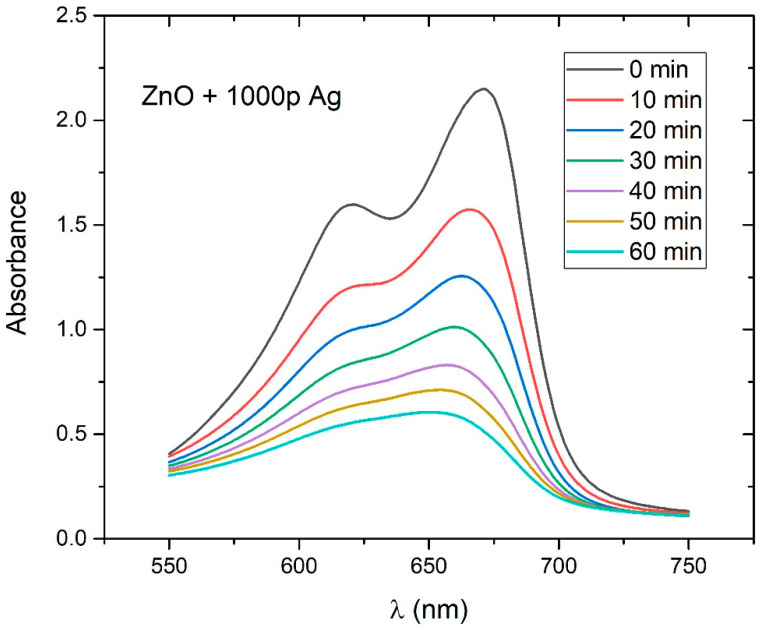
Photocatalytic degradation of MB in PLAL-synthesized colloidal solution of Ag-doped ZnO NPs (1000p used for Ag deposition in PLD) under UV irradiation.

**Figure 12 nanomaterials-12-02677-f012:**
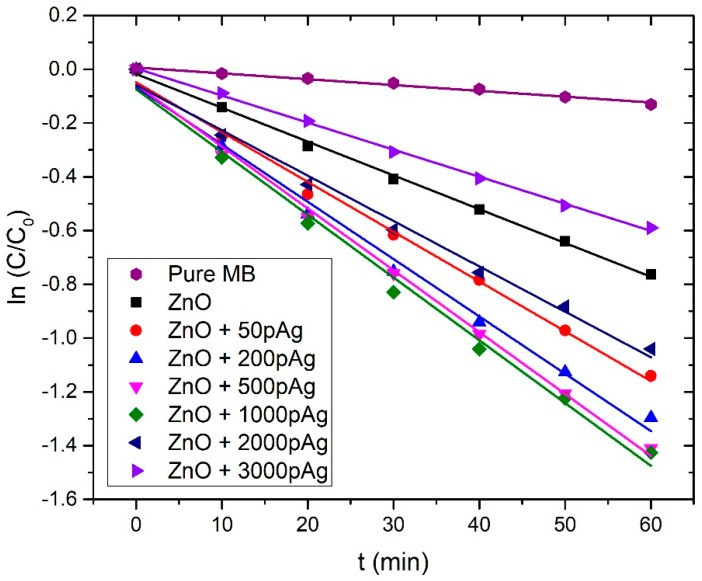
ln(*C*/*C*_0_) as a function of time for photocatalytic degradation of MB in different ZnO NP colloidal solutions under UV irradiation.

**Figure 13 nanomaterials-12-02677-f013:**
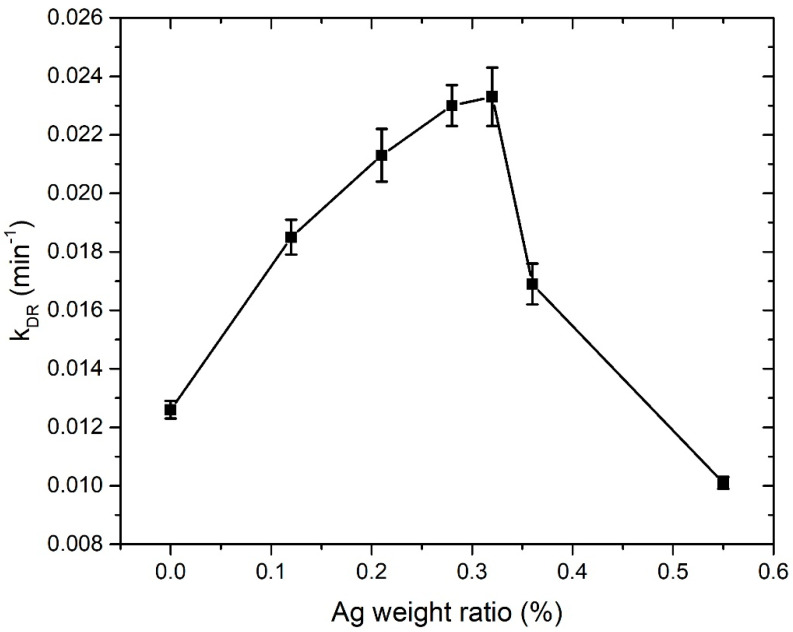
Photodegradation rate dependence on ICP-OES calculated Ag weight ratio in Ag-doped ZnO NPs.

**Table 1 nanomaterials-12-02677-t001:** Lattice constants a and c for pure and Ag-doped ZnO NPs.

Lattice Constant (Orientation)	Pure ZnO	Ag-Doped ZnO
a (100)/Å	3.23	3.25
c (002)/Å	5.17	5.21

**Table 2 nanomaterials-12-02677-t002:** MB photodegradation rates and photodegradation half-times under UV irradiation, derived from pseudo-first order reaction model for different Ag weight ratios in colloidal ZnO catalyst.

Solution	Ag Weight Ratio (ICP-OES)/%	*k_DR_*/min^−1^	*t*_1/2_/min
Pure MB	/	0.00217	319.4
ZnO	0	0.0126	55.0
ZnO_50Ag	0.12	0.0185	37.5
ZnO_200Ag	0.21	0.0213	32.5
ZnO_500Ag	0.28	0.0230	30.1
ZnO_1000Ag	0.32	0.0233	29.7
ZnO_2000Ag	0.36	0.0169	41.0
ZnO_3000Ag	0.55	0.0101	68.6

## Data Availability

Not applicable.

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
