# Peer review of "Enhancement of Methylene Blue Photodegradation Rate Using Laser Synthesized Ag-Doped ZnO Nanoparticles"

_nanomaterials, 2022, doi:10.3390/nano12152677_

Round 1

Reviewer 1 Report

Krstulović et al prepared Ag-doped ZnO nanoparticles by pulsed laser ablation of the Ag coated ZnO target. The nanoparticles were characterized by various experimental methods, and the photocatalytic performance in the degradation of Methylene Blue (MB). Some expressions are not proper, required to be promoted. I consider it for acceptance if the followed concerns are addressed:

1.        In addition, the significance and innovation of this work is not clearly, the introduction section should be rewritten.  

2.        Recently, ZnO based nanomaterials have been widely applied in industry for the production of transistors and detectors. You’d better added some relative references in the trodcuction (for example: A Visible-Blind Photodetector and Artificial Optoelectronic Synapse Using Liquid-Metal Exfoliated ZnO Nanosheets, ADVANCED OPTICAL MATERIALS, 2021, 9(16), 2100449; ZnO Nanofiber Thin-Film Transistors with Low-Operating Voltages, ADVANCED ELECTRONIC MATERIALS, 2018, 4(1), 1700336; Effective encapsulation of ZnO thin film transistors controlled by thermal energy, APPLIED SURFACE SCIENCE, 2021, 548, 149253; Combustion Synthesized Electrospun InZnO Nanowires for Ultraviolet Photodetectors, ADVANCED ELECTRONIC MATERIALS, 2021, 8, 4, 2100997.)

3.        In Figure 2, the clear scale bars should be provided.

4.        All the formulas in this article should be aligned.

5.        The TEM and HRTEM pictures should be provided to exhibit the microstructure of the Ag-doped ZnO nanoparticles, and then discuss the structure-property relationship.

6.        The authorship should further explain why ZnO NP doped with 0.32 wt % silver exhibiting the largest photocatalytic efficiency.

7.        The catalytic mechanism of Ag-doped ZnO nanoparticles on MB should be explained and proved.

Author Response

Reviewer #1:

Krstulović et al prepared Ag-doped ZnO nanoparticles by pulsed laser ablation of the Ag coated ZnO target. The nanoparticles were characterized by various experimental methods, and the photocatalytic performance in the degradation of Methylene Blue (MB). Some expressions are not proper, required to be promoted. I consider it for acceptance if the followed concerns are addressed:

Question 1:

  1. In addition, the significance and innovation of this work is not clearly, the introduction section should be rewritten.

Answer 1:

  1. The introduction section is modified. The innovation of this work is now outlined in introduction. Changes are labeled in red. The significance of this work is that we propose a new method for syntesis of Ag-doped ZnO nanoparticles and application in photocatalysis of organic dyes.

Question 2:

  1. Recently, ZnO based nanomaterials have been widely applied in industry for the production of transistors and detectors. You’d better added some relative references in the trodcuction (for example: A Visible-Blind Photodetector and Artificial Optoelectronic Synapse Using Liquid-Metal Exfoliated ZnO Nanosheets, ADVANCED OPTICAL MATERIALS, 2021, 9(16), 2100449; ZnO Nanofiber Thin-Film Transistors with Low-Operating Voltages, ADVANCED ELECTRONIC MATERIALS, 2018, 4(1), 1700336; Effective encapsulation of ZnO thin film transistors controlled by thermal energy, APPLIED SURFACE SCIENCE, 2021, 548, 149253; Combustion Synthesized Electrospun InZnO Nanowires for Ultraviolet Photodetectors, ADVANCED ELECTRONIC MATERIALS, 2021, 8, 4, 2100997.)

Answer 2:

  1. Two references are added in Introduction section (ref 4 and 5).

Question 3:

  1. In Figure 2, the clear scale bars should be provided.

Answer 3:

  1. Figure 2 is modified and the scale bar is now clear.

Question 4:

  1. All the formulas in this article should be aligned.

Answer 4:

  1. All formulae will be aligned after editing.

Question 5:

  1. The TEM and HRTEM pictures should be provided to exhibit the microstructure of the Ag-doped ZnO nanoparticles, and then discuss the structure-property relationship.

Answer 5:

  1. The TEM images of pure ZnO and Ag-doped ZnO NPs are shown in a new Figure 3 with description and discussion marked in red.

''In Figure 3, a TEM images for pure (a, b) and Ag-doped (c, d) ZnO NP are presented. Fig-ure 3 a) and c) shows presence of spherical nanoparticles with different sizes and irregu-lar material formed during LAL process. In Figure 3 b) and d) show surface morphology of nanoparticles. It can be seen that in both cases NP surface is relatively smooth. Neverthe-less, the edges are not perfectly sharp indicating that NP surface has some roughness. The differences in photocatalytic activities between pure and Ag-doped ZnO NP are ob-viously not dependent on surface morphology.''   

Question 6:

  1. The authorship should further explain why ZnO NP doped with 0.32 wt % silver exhibiting the largest photocatalytic efficiency.

Answer 6:

  1. The optimized Ag weight ratio is now explained in a little bit more details but still qualitatively. The added text is marked in red (in section 3.7, paragraph before last one).

Question 7:

  1. The catalytic mechanism of Ag-doped ZnO nanoparticles on MB should be explained and proved.

Answer 7:

  1. We added the possible photocatalytic mechanism in the section 3.7. marked in red:

''Photocatalytic activity of Ag-doped ZnO NPs is based on the excitation of the electrons from the valence band (VB) to the conduction band (CB) under light irradiation that is equal or higher than band-gap energy. When electrons are excited, holes are formed in the VB that act as positive charge carriers. Unique Ag/ZnO Fermi level is below the bottom of the ZnO CB, so photo-excited electrons flow from ZnO to the Ag surface (Ag is an “electron sink”) while the holes remain on the ZnO surface. Moreover, silver doping can increase photocatalytic activity by creating band levels that act as traps for charge carriers. Therefore, the recombination rate of electrons and holes decreases, and their lifetime increases [12]. New band levels can also lead to effective band-gap narrowing that expands the range of photon energies, which can lead to photocatalytic efficiency in the visible part of the spectrum [10]. Silver, as plasmonic metal, strongly enhances the light absorption via surface plasmon resonance (SPR), increasing the number of excited electrons and holes in the catalyst, consequently leading to higher photocatalytic performance [10]. Ag usually has a higher value of work function (4.3 eV [16]) than the absolute value of ZnO electron affinity (about 4.1 eV [17]). When the balance between the Fermi levels is established, the Schottky barrier is formed at ZnO/Ag [18]. Schottky junction is the depleted area where a strong electric field separates the electrons and holes. This process decreases the e-/h+ recombination rate and, therefore, improves photocatalysis. In photocatalytic reaction, electrons in the CB react with O2 generating superoxide anion (.O2-), while holes in the VB react with H2O generating hydroxyl radical (.OH). Therefore, these free radicals participate in the photodegradation of organic dyes, such as MB [1-3]. In [3], Ag-doped ZnO nanocatalysts were synthesized using the co-precipitation method. Intriguingly, a scavenger test was done to determine the radicals that affect MB photodegradation. Results showed that photo-generated holes and .OH radicals are the main species, while .O2- radicals are supportive species in the photodegradation process of the MB.''

Reviewer 2 Report

The manuscript “Enhancement of Methylene Blue Photodegradation Rate Using Laser Synthesized Ag-Doped ZnO Nanoparticles” reports about the fabrication of Ag-doped ZnO nanoparticles obtained by means of laser ablation technique. The nanostructures were characterized by XRD, XPS, SEM, EDS, ICP-OES and UV-VIS spectrophotometry and their photocatalytic activity when irradiated under UV photons was evaluated using methylene blue as model compound. A large part of the manuscript is dedicated to the chemical physical characterization of the Ag-ZnO NPs and the effect of Ag both on the electronic and crystal structure of ZnO was deeply studied. In my opinion, there are 2 points that needs to be clarified by the authors:

1. SEM characterization appears confused: the morphologies of PLAL-synthesized Ag-doped ZnO NP using 1000p and 500p are reported in figure 2a and 2b, but in figure 2c ZnO size-distribution of ZnO NP is reported. Why the size-distribution of Ag-doped ZnO NP 1000p and Ag-doped ZnO NP 500p and SEM images of bare ZnO are not reported?

Furthermore, in the experimental sections related to XPS, ICP and photodegradation structures obtained by different laser pulses are reported. On the contrary, in the section “SEM images” the authors report only 500p and 1000p, without any explanation. This point should be briefly clarified and the section dedicated to morphological characterization should be renamed.

2.       The photocatalytic action of the catalysts is neglected. No rationalization about the mechanism that induced the MB photodegradation is reported: is it an electron mediated mechanism? Or is it ruled by the hydroxyl radical photo-production? Experiments using scavengers could be helpful for the authors to clarify this point.

Does the Ag doping influence the spectral range where the photodegradation takes place?

Author Response

Reviewer #2:

The manuscript “Enhancement of Methylene Blue Photodegradation Rate Using Laser Synthesized Ag-Doped ZnO Nanoparticles” reports about the fabrication of Ag-doped ZnO nanoparticles obtained by means of laser ablation technique. The nanostructures were characterized by XRD, XPS, SEM, EDS, ICP-OES and UV-VIS spectrophotometry and their photocatalytic activity when irradiated under UV photons was evaluated using methylene blue as model compound. A large part of the manuscript is dedicated to the chemical physical characterization of the Ag-ZnO NPs and the effect of Ag both on the electronic and crystal structure of ZnO was deeply studied. In my opinion, there are 2 points that needs to be clarified by the authors:

Question 1:

  1. SEM characterization appears confused: the morphologies of PLAL-synthesized Ag-doped ZnO NP using 1000p and 500p are reported in figure 2a and 2b, but in figure 2c ZnO size-distribution of ZnO NP is reported. Why the size-distribution of Ag-doped ZnO NP 1000p and Ag-doped ZnO NP 500p and SEM images of bare ZnO are not reported?

Furthermore, in the experimental sections related to XPS, ICP and photodegradation structures obtained by different laser pulses are reported. On the contrary, in the section “SEM images” the authors report only 500p and 1000p, without any explanation. This point should be briefly clarified and the section dedicated to morphological characterization should be renamed.

Answer 1:

  1. We changed Figure 2 (SEM images) and added a new Figure 3 (TEM images). In Figure 2 we show SEM image of pure and Ag-doped ZnO with corresponding size-distributions. Size distributions and morphologies are independent on the number of applied laser pulses (including pure ZnO). We clarified that in text marked in red. From Figure 3 it can be seen that pure and Ag.doped ZnO NP have the same surface morphology.

Question 2:

  1. The photocatalytic action of the catalysts is neglected. No rationalization about the mechanism that induced the MB photodegradation is reported: is it an electron mediated mechanism? Or is it ruled by the hydroxyl radical photo-production? Experiments using scavengers could be helpful for the authors to clarify this point.

Does the Ag doping influence the spectral range where the photodegradation takes place?

Answer 2:

  1. Unfortunately, we are unable to do a scavenger test at this time. However, we have investigated the literature, and, below, we explained the most likely photodegradation mechanism of Ag-doped ZnO in the presence of Methylene Blue (MB). Also, we cite an article in which an Ag-doped ZnO nanocatalyst performed a scavenger test, and probably the same radicals dominate in our photodegradation of MB. Of course, in the future, we plan to do scavenger tests for photocatalytic reactions. The explanation is added in text and marked in red:

''Photocatalytic activity of Ag-doped ZnO NPs is based on the excitation of the electrons from the valence band (VB) to the conduction band (CB) under light irradiation that is equal or higher than band-gap energy. When electrons are excited, holes are formed in the VB that act as positive charge carriers. Unique Ag/ZnO Fermi level is below the bottom of the ZnO CB, so photo-excited electrons flow from ZnO to the Ag surface (Ag is an “electron sink”) while the holes remain on the ZnO surface. Moreover, silver doping can increase photocatalytic activity by creating band levels that act as traps for charge carriers. Therefore, the recombination rate of electrons and holes decreases, and their lifetime increases [14]. New band levels can also lead to effective band-gap narrowing that expands the range of photon energies, which can lead to photocatalytic efficiency in the visible part of the spectrum [12]. Silver, as plasmonic metal, strongly enhances the light absorption via surface plasmon resonance (SPR), increasing the number of excited electrons and holes in the catalyst, consequently leading to higher photocatalytic performance [12]. Ag usually has a higher value of work function (4.3 eV [18]) than the absolute value of ZnO electron affinity (about 4.1 eV [19]). When the balance between the Fermi levels is established, the Schottky barrier is formed at ZnO/Ag [20]. Schottky junction is the depleted area where a strong electric field separates the electrons and holes. This process decreases the e-/h+ recombination rate and, therefore, improves photocatalysis. In photocatalytic reaction, electrons in the CB react with O2 generating superoxide anion (.O2-), while holes in the VB react with H2O generating hydroxyl radical (.OH). Therefore, these free radicals participate in the photodegradation of organic dyes, such as MB [51,52,53]. In [53], Ag-doped ZnO nanocatalysts were synthesized using the co-precipitation method. Intriguingly, a scavenger test was done to determine the radicals that affect MB photodegradation. Results showed that photo-generated holes and .OH radicals are the main species, while .O2- radicals are supportive species in the photodegradation process of the MB. We assume that the same radicals dominates in our case as the crystallinity of nanoparticles and Ag weight ratios are similar.''

Ag doping influence the UV-Vis spectrum as it can be seen from Figure 10. The decrease in Band-gap energy is observed and increase in visible absorption.

Reviewer 3 Report

This manuscript entitled ‘Enhancement of Methylene Blue Photodegradation Rate Using  Laser Synthesized Ag-Doped ZnO Nanoparticles’ describes the pulsed laser ablation of the Ag coated ZnO target in water is performed in order to synthesize the colloidal solution of Ag-doped ZnO nanoparticles. The synthesized materials were well characterized by various methods in order to determine their crystal and stoichiometric structure, size distribution, shape, optical properties and mass concentration. Moreover, the authors successfully applied the dopant materials used as a catalyst in photocatalytic degradation of Methylene Blue. Thus, this reviewer recommends this work for publication in the Nanomaterials Journal after addressing the following minor comments:

Comments:

The author should re-write the abstract section. It showed fully self-plagiarized. Also it should be contain some quantitative information.

The author should provide the TEM morphological measurements with particle size distribution of the Ag-doped ZnO nanoparticles.

The author should propose the possible mechanism of Ag-doped ZnO nanoparticles in the degradation of Methylene Blue in detail.

The manuscript contains some typographical, grammatical errors and superfluous spaces that need to be corrected throughout the manuscript.

Author Response

Reviewer #3:

This manuscript entitled ‘Enhancement of Methylene Blue Photodegradation Rate Using  Laser Synthesized Ag-Doped ZnO Nanoparticles’ describes the pulsed laser ablation of the Ag coated ZnO target in water is performed in order to synthesize the colloidal solution of Ag-doped ZnO nanoparticles. The synthesized materials were well characterized by various methods in order to determine their crystal and stoichiometric structure, size distribution, shape, optical properties and mass concentration. Moreover, the authors successfully applied the dopant materials used as a catalyst in photocatalytic degradation of Methylene Blue. Thus, this reviewer recommends this work for publication in the Nanomaterials Journal after addressing the following minor comments:

Question 1:

  1. The author should re-write the abstract section. It showed fully self-plagiarized. Also it should be contain some quantitative information.

Answer 1:

  1. The abstract is rewritten and contains some quantitative information.

Question 2:

  1. The author should provide the TEM morphological measurements with particle size distribution of the Ag-doped ZnO nanoparticles.

Answer 2:

  1. The TEM images of pure ZnO and Ag-doped ZnO NPs are shown in a new Figure 3 with description and discussion marked in red.

''In Figure 3, a TEM images for pure (a, b) and Ag-doped (c, d) ZnO NP are presented. Fig-ure 3 a) and c) shows presence of spherical nanoparticles with different sizes and irregu-lar material formed during LAL process. In Figure 3 b) and d) show surface morphology of nanoparticles. It can be seen that in both cases NP surface is relatively smooth. Neverthe-less, the edges are not perfectly sharp indicating that NP surface has some roughness. The differences in photocatalytic activities between pure and Ag-doped ZnO NP are ob-viously not dependent on surface morphology.''   

Size distribution of pure and Ag-doped ZnO NP is shown in modified Figure 2 (SEM images).

Question 3:

  1. The author should propose the possible mechanism of Ag-doped ZnO nanoparticles in the degradation of Methylene Blue in detail.

Answer 3:

  1. An explanation of the photodegradation mechanism of methylene blue in the presence of Ag-doped ZnO was added in the article (Results section, text marked in red) and is given below:

''Photocatalytic activity of Ag-doped ZnO NPs is based on the excitation of the electrons from the valence band (VB) to the conduction band (CB) under light irradiation that is equal or higher than band-gap energy. When electrons are excited, holes are formed in the VB that act as positive charge carriers. Unique Ag/ZnO Fermi level is below the bottom of the ZnO CB, so photo-excited electrons flow from ZnO to the Ag surface (Ag is an “electron sink”) while the holes remain on the ZnO surface. Moreover, silver doping can increase photocatalytic activity by creating band levels that act as traps for charge carriers. Therefore, the recombination rate of electrons and holes decreases, and their lifetime increases [14]. New band levels can also lead to effective band-gap narrowing that expands the range of photon energies, which can lead to photocatalytic efficiency in the visible part of the spectrum [12]. Silver, as plasmonic metal, strongly enhances the light absorption via surface plasmon resonance (SPR), increasing the number of excited electrons and holes in the catalyst, consequently leading to higher photocatalytic performance [12]. Ag usually has a higher value of work function (4.3 eV [18]) than the absolute value of ZnO electron affinity (about 4.1 eV [19]). When the balance between the Fermi levels is established, the Schottky barrier is formed at ZnO/Ag [20]. Schottky junction is the depleted area where a strong electric field separates the electrons and holes. This process decreases the e-/h+ recombination rate and, therefore, improves photocatalysis. In photocatalytic reaction, electrons in the CB react with O2 generating superoxide anion (.O2-), while holes in the VB react with H2O generating hydroxyl radical (.OH). Therefore, these free radicals participate in the photodegradation of organic dyes, such as MB [51,52,53]. In [53], Ag-doped ZnO nanocatalysts were synthesized using the co-precipitation method. Intriguingly, a scavenger test was done to determine the radicals that affect MB photodegradation. Results showed that photo-generated holes and .OH radicals are the main species, while .O2- radicals are supportive species in the photodegradation process of the MB. We assume that the same radicals dominates in our case as the crystallinity of nanoparticles and Ag weight ratios are similar.''

Question 4:

  1. The manuscript contains some typographical, grammatical errors and superfluous spaces that need to be corrected throughout the manuscript.

Answer 4:

  1. Corrections have been made throughout the article.

Round 2

Reviewer 1 Report

The authors have replied all the questions and made relative revision.

Reviewer 2 Report

Blažeka and co-workers adequately replied to the proposed revisions and the manuscript is now suitable for publication.